# The Impact of Revascularization in a Patient with Atypical Manifestations of Hypoperfusion

**DOI:** 10.3390/medicina58101328

**Published:** 2022-09-22

**Authors:** Sintija Strautmane, Zanda Priede, Andrejs Millers

**Affiliations:** 1Faculty of Residency, Riga Stradins University, Dzirciema Iela 16, Riga LV—1007, Latvia; 2Department of Neurology, Pauls Stradins Clinical University Hospital, Pilsonu Iela 13, Riga LV—1002, Latvia; 3Department of Neurology and Neurosurgery, Riga Stradins University, Dzirciema Iela 16, Riga LV—1007, Latvia

**Keywords:** revascularization, transient ischemic attack, hypoperfusion, ischemic stroke, internal carotid artery, subocclusion, stenosis, hyperkinesis, clinical manifestations

## Abstract

*Background and Objectives*: Carotid revascularization is one of the most effective treatment options in patients with severe carotid artery stenosis causing hypoperfusion in basal ganglia. Atypical manifestations include hyperkinetic movements, noted as extremely rare. We report a case about a patient with 2-months-long complaints of Uncontrollable movements in his right side of the body subsided after carotid revascularization. *Case presentation*: A 71-year-old male was admitted to Pauls Stradins Clinical University Hospital with the main complaints of 2-months-long uncontrollable movements in his right hand and his right leg. When performing coordination tasks, slight inaccuracy was noted with the right-side extremities. Hyperkinetic movements—choreoathetosis in the right side of the patient’s face, arm, and leg—were seen. Computed tomography angiography revealed subocclusion in the proximal segment of the left internal carotid artery and 30% stenosis in the proximal segment of the right internal carotid artery. The patient was consulted by a vascular surgeon. Eversion endarterectomy of the left internal carotid artery was performed. The early postoperative period occurred without complications. The patient was discharged from the hospital 2 days after the surgery in good overall health condition. Two months later, choreoathetotic movements in his right side of the body had markedly decreased. No focal neurologic deficits were noted. *Conclusions*: Revascularization may be effective by eliminating emboli and stenosis, leading to hypoperfusion in watershed territories. A case of a 71-year-old male patient with the main complaints of 2-months-long uncontrollable movements in his right side of the body subsiding after carotid revascularization was demonstrated. It is vital to recognize atypical manifestations of hypoperfusion, associated with stenosis in internal carotid arteries, to early make a diagnosis, to perform an appropriate treatment, and to reduce the risk of cerebral infarction in the future, resulting in a longer high-quality life for the patient.

## 1. Introduction

According to the American Heart Association/American Stroke Association guidelines for the prevention of stroke in patients with ischemic stroke and transient ischemic attack, published in 2021, extracranial carotid artery disease is an important and treatable cause of ischemic stroke.

Atypical manifestations of severe extracranial carotid artery stenosis, near-total or total occlusion cause hypoperfusion in basal ganglia that receive perfusion mainly from branches of the internal carotid artery, especially in the striatum, as well as in the caudate nucleus, putamen, thalamus, subthalamic nucleus, subcortical white matter of the brain [1,2,3,4]. Hypoperfusion in basal ganglia may result in hyperkinetic movements, for instance, hemichorea, athetosis, hemiballism, etc., noted extremely rare [1,3,4].

Carotid artery stenosis as one of the differential diagnoses of the causes of chorea is reported just recently. It is of massive importance not only to be competent in hyperkinetic movements but also to observe patients very carefully [1,3]. 

Other diseases that may cause hypoperfusion resulting in atypical clinical manifestations include Moya–moya disease. In this disease, there is reduced activation of the indirect pathway of the corticobasal ganglia—thalamocortical—loop leading to a hypermetabolic state of the striatum and cortex that eventually leads to increased activation of the excitatory neurons [5]. Cortical and subcortical hypoperfusion in frontoparietal and temporal occipital regions also may be causes of atypical manifestations of hypoperfusion [6]. 

Carotid revascularization is one of the most effective treatment options in patients with severe carotid artery stenosis. Carotid revascularization options include carotid artery endarterectomy (CEA) and carotid artery stenting (CAS). CEA is established as safe and effective, but CAS may be useful in patients who are not candidates for CEA. Regardless of the technique, the perioperative risk of ischemic stroke and mortality for the surgeon or the center should be less than 6% to provide an overall benefit for the patient [7,8,9]. 

Till now, in Pauls Stradins Clinical University Hospital, 4 cases have been reported demonstrating the rarity and variability of hypoperfusion caused by extracranial magistral blood vessel stenosis. 

In this manuscript, we demonstrate a clinical case of a patient who was a 71-year-old male with 2-months-long complaints of uncontrollable movements in his right hand and his right leg subsiding after a carotid revascularization procedure. 

## 2. Case Presentation

A 71-year-old male was admitted to Pauls Stradins Clinical University Hospital with complaints of uncontrollable movements in his right hand and right leg. When asked, the patient revealed these movements had been ongoing for 2 months.

Patient comorbidities included coronary heart disease, history of myocardial infarctions, PTCA RCA and LAD stents (2008), CRT-D (2009, 2016), chronic heart failure, class II (NYHA) with a left ventricular ejection fraction of 30%. According to the patient, no history of previous stroke or other medical conditions that cause limb tremors were noted. 

During a neurological examination, the patient was alert, and sightly non-critical with a little dysphonic speech that the patient noted had not changed in any way. Pupils were symmetrical, dx = sin with the reaction to light retained. Eyeball movements in a full range, patient’s face symmetrical. When asked to show his tongue, involuntary movements in the tongue were noted with the tongue situated centrally. Next, the patient was able to move both his upper and lower extremities, therefore, no paresis was observed. When asked to perform coordination tasks, slight inaccuracy was noted with the patient’s extremities of the right side, while with his left arm and left leg, the task was performed precisely. During the examination, uncontrollable movements—choreoathetosis in the right side of the patient’s face, arm, and leg were seen. The patient demonstrated slight postural tremor in his upper extremities, sin > dx. When performing Romberg’s test, the patient was able to stand still but admitted feeling unstable. No alterations in sensation were observed. 

No significant changes were noted in the patient blood tests and patient urine analysis. When performed computed tomography angiography (CTA), subocclusion in the proximal segment of the left internal carotid artery was seen, as well as 30% stenosis in the proximal segment of the right internal carotid artery was observed (see Figure 1 and Figure 2).

Ultrasonography of the thyroid gland and neck showed a little cyst and benign nodus in the left lobe of the thyroid gland with TIRADS 2. 

Brain magnetic resonance imaging (MRI) with magnetic resonance angiography (MRA) with intravenous contrast was performed with sequences as follows: FLAIR, MP reconstructions, T2 SE ax, DW ax, SW ax, MRA, 3D TOF, and MIP reconstructions, T2 core with fat saturation, T1 SE sag post-contrast. The radiological findings included diffusely expanded ventricles and cortical grooves, as well as segmentary stenosis in the extracranial part of the left internal carotid artery (see Figure 3). 

In the Neurology Department, the patient received antiaggregant therapy with Acetylsalicylic acid as well as intensive therapy with statins and antihypertensive medication. Due to the subocclusion in the proximal segment of the left internal carotid artery, the patient was consulted by a vascular surgeon. Surgical revascularization of the subocclusion was indicated. 

The patient was transferred to the vascular surgery department with the main diagnosis of atherosclerosis in brachiocephalic blood vessels—symptomatic subocclusion of the left internal carotid artery with choreoathetosis in the right sight of the body and with stenosis of 30% in the proximal segment of the right internal carotid artery. Next, patient blood tests were repeated with no significant alterations. Lung computed tomography demonstrated no pathology in the lungs. With suspicion of malignancy, computed tomography of the abdomen and pelvis with intravenous contrast was performed—no data of tumor were detected. 

Eversion endarterectomy of the left internal carotid artery was performed. The early postoperative period occurred without complications. The wound healed primarily. The patient was discharged from the hospital 2 days after the surgery in good overall health condition. 

Two months later, the patient saw a neurologist in a control appointment. The patient was overall healthy, demonstrated no complaints, and noted that choreoathetotic movements in his right side of the body had markedly decreased. No focal neurologic deficits were noted. The patient main diagnoses at the control appointment included eversion endarterectomy of the left internal carotid artery with right side choreoathetosis due to subocclusion of the left internal carotid artery.

## 3. Discussion

Carotid revascularization, including carotid artery endarterectomy and carotid artery stenting, is one of the most effective treatment options in patients with severe carotid artery stenosis. 

Atypical manifestations of severe extracranial carotid artery stenosis, near-total or total occlusion, causing hypoperfusion in basal ganglia include hyperkinetic movements, for instance, hemichorea, athetosis, hemiballism, etc. According to the literature, these manifestations are noted as extremely rare.

There are few published series examining the vascular causes of new-onset chorea. For an instance, an article about the resolution of hemichorea in three patients following endarterectomy for severe carotid stenosis was published in 2008 (see Table 1) [4]. All three patients were more than 70 years old, two males and one female, and the duration of their hyperkinetic movements (chorea) before the carotid artery endarterectomy were from 1 month to 17 months. One of the patients reported a subacute onset of the course while the other two reported acute onset of the course. All patients had severe left internal carotid artery stenosis; therefore, the chorea was noted on the right side of their bodies. Nevertheless, all patients demonstrated a full recovery of chorea after carotid endarterectomy: immediate recovery was noted in one patient, another patient recovered 2 weeks after the procedure, but the lady demonstrated recovery 6 months after the procedure. All the patients were followed up from 1 year to 5 years after carotid endarterectomy [4].

Extracranial carotid artery disease is an important and treatable cause of ischemic stroke. Brain and neck visual diagnostics should be performed in all patients with atypical manifestations of hypoperfusion including chorea regardless of patient symptoms. 

Revascularization may be effective by the elimination of emboli and stenosis that leads to hypoperfusion in watershed territories. It is known that there are two main surgical techniques for carotid endarterectomy: the classical or conventional method and the eversion method. 

In the classical or conventional method, the surgeon makes an incision alongside the medial aspect of the sternocleidomastoid muscle, exposing the carotid arteries. The internal carotid artery is clamped proximal and distal to the plaque temporarily stopping blood flow. Eventually, the clamps are released one at a time, and a flexible bypass stent is placed. Next, the artery is then repaired with a patch widening the vessel lumen, and the bypass stent is removed [10]. In cases when carotid plaque is extended more than usual, as well as, if the usage of the external shunt is necessary, the conventional method is preferred. 

In this case, an eversion carotid endarterectomy was performed. This is known not only as the main procedure in carotid surgery, but also it is the most frequent vascular procedure [11]. 

With this method, the internal carotid artery is transected at its origin at the bifurcation. The vessel wall is everted circumferentially around the plaque, and the plaque is divided and removed. The artery is then repaired in an end-to-end anastomotic fashion. Therefore, the blood flow is improved resulting in a reduction of involuntary movements [10,12]. The advantages of this method include no need for patch closure and the overall carotid clamping time being shorter, therefore, the total ischemic and operative time is also shorter. Due to the oblique shape of anastomosis, eversion carotid endarterectomy is associated with a low risk of long-term restenosis [11]. 

In our case, the patient was a 71-year-old male whose main complaints included 2-months-long uncontrollable movements in his right hand and his right leg. Similarly, as the patients demonstrated in the article published in 2008, our patient underwent an eversion endarterectomy of the left internal carotid artery. After the procedure, the patient uncontrollable movements subsided rapidly. 

It is vital to recognize atypical manifestations of hypoperfusion, including hyperkinetic movements, such as hemichorea, associated with stenosis in internal carotid arteries, to early make a diagnosis, perform an appropriate treatment, and reduce patient risk of cerebral infarction in the future. 

Carotid artery stenosis should be considered in the differential diagnosis of unilateral hyperkinetic movements, even in the absence of preceding presentation with ischemic stroke or transient ischemic attacks. 

Further research is warranted to assess factors contributing to atypical manifestations of hypoperfusion.

## 4. Conclusions

Extracranial carotid artery disease is an important and treatable cause of ischemic stroke, and carotid revascularization is one of the most effective treatment options to reduce patient risk of cerebral infarction in the future. 

Atypical manifestations of severe extracranial carotid artery stenosis noted as extremely rare, include hyperkinetic movements. Its several causes include metabolic, vascular, structural, autoimmune, and genetic. It is extremely important not only to be competent in hyperkinetic movements but also to observe patients very carefully. 

This case report marks the importance of detailed patient complaints assessment when diagnosing atypical manifestations of hypoperfusion. This case report also proves the possibility to diagnose a rare cause of hyperkinetic movements.

It is possible to lead a normal, high-quality life after successful revascularization in a critical carotid artery stenosis. 

## Figures and Tables

**Figure 1 medicina-58-01328-f001:**
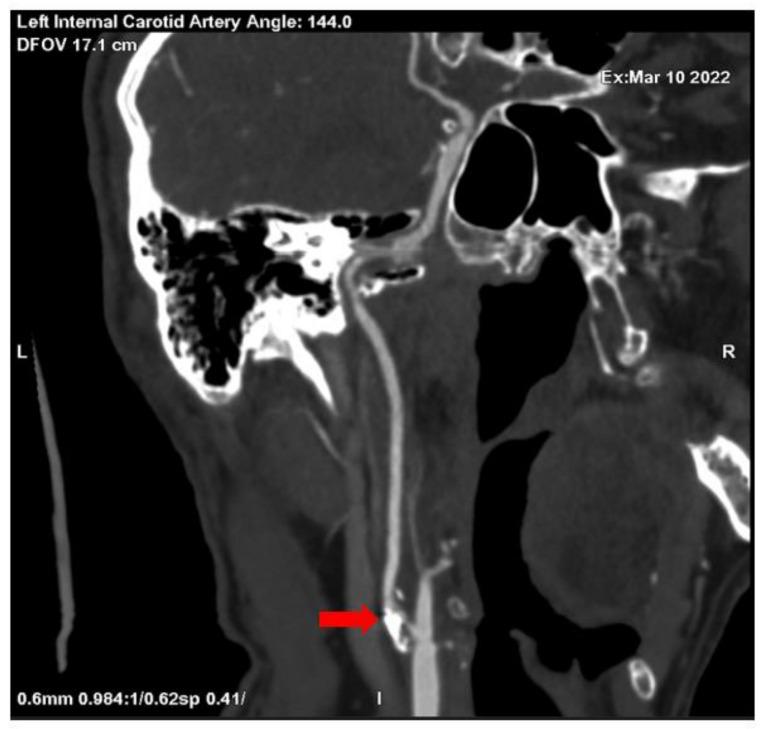
Subocclusion in the proximal segment of the left internal carotid artery (CTA) (red array).

**Figure 2 medicina-58-01328-f002:**
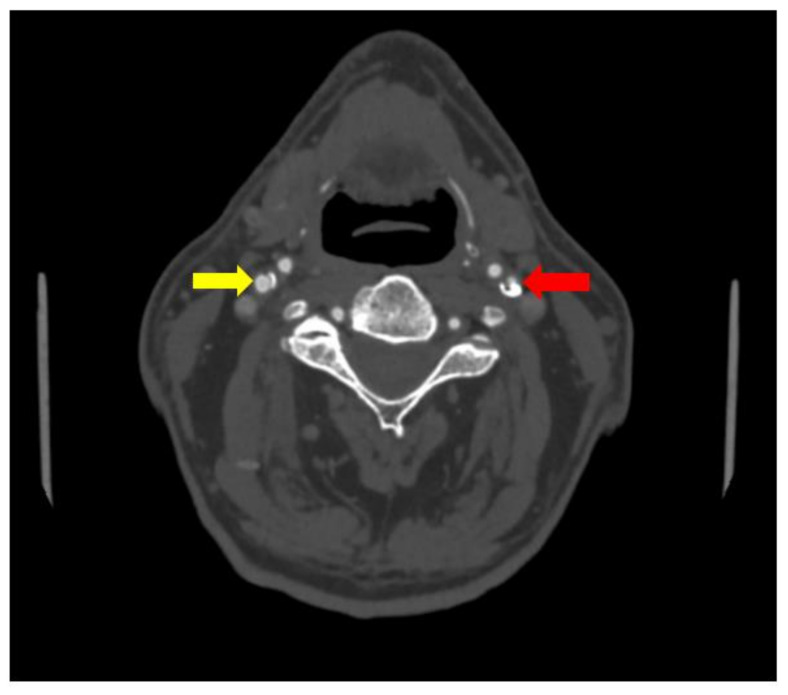
Subocclusion in the proximal segment of the left internal carotid artery (red array) and 30% stenosis in the proximal segment of the right internal carotid artery (CTA) (yellow array).

**Figure 3 medicina-58-01328-f003:**
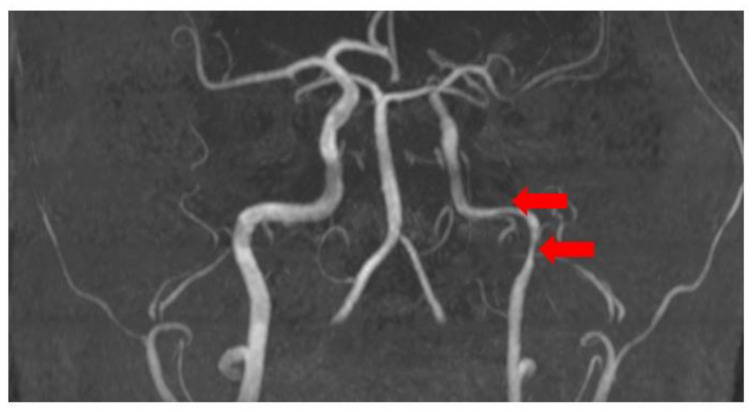
Segmentary stenosis in the extracranial part of the left internal carotid artery (MRA) (red arrays).

**Table 1 medicina-58-01328-t001:** Resolution of hemichorea following endarterectomy for severe carotid stenosis (2008) [4].

	Patient A	Patient B	Patient C
Age	72 years	75 years	73 years
Gender	Female	Male	Male
Semiology of the chorea at the onset	Right arm > leg	Right arm & leg	Right arm & leg
Duration of chorea before CEA	17 months	3 months	1 month
Time course of onset	Acute	Acute	Subacute
Medication at presentation	Aspirin, ramipril, simvastatin	Aspirin, nebivolol, lansoprazole	Thyroxine, nifedipine
Family history of movement disorders	None	None	None
Other neurological signs	None	Right cortical sensory loss, right UMN weakness face & hand	Right cortical sensory loss
Carotid bruit	None	None	Left carotid bruit
Blood test panel	All normal	All normal	All normal
CVD risk factors	IHD, HT, PVD, smoker	HT, ex-smoker, FH	HT, FH
Carotid stenosis on duplex ultrasonography	90% left ICA, 25–50% right ICA	90% left ICA, <25% right ICA	90% left ICA, <25% right ICA
Brain imaging	CT normal	MRI: left posterior parietal gray matter infarct	MRI: left anterior parietal lobe infarct
Response to antichoreic agent	Partial to haloperidol	Not tried	Not tried
Recovery of chorea after carotid endarterectomy	Full	Full	Full
Time to resolution of chorea	6 months	2 weeks	Immediately
Follow-up period since carotid endarterectomy	4 years	5 years	1 year

Abbreviations: CEA—carotid endarterectomy; UMN—upper motor neuron; CVD: cerebrovascular disease; IHD—ischemic heart diseases; HT—hypertension; PVD—peripheral vascular disease; FH—family history; ICA—internal carotid artery.

## Data Availability

The data presented in this study are available on request from the corresponding author.

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
