# Peer review of "The Impact of Revascularization in a Patient with Atypical Manifestations of Hypoperfusion"

_medicina, 2022, doi:10.3390/medicina58101328_

Round 1

Reviewer 1 Report

The author explained that the involuntary limb movement caused by carotid artery stenosis can be effectively eliminated by carotid artery revascularization. But this topic seems to have been reported and elaborated by some researchers in the past decade or so. Therefore, I believe that there is not much new to learn in this article. Here are my comments:

First of all, there are some grammatical errors and some long sentences with semantic deviation in the article. It is suggested that the author use multiple short sentences to describe the content or make professional English polishing.

Abstract:

1.  Suggested that the "We report a case about a patient with 2 - up - long complaints of uncontrollable movements in his right side of the Body subsiding after carotid revascularization."(L.18-20) To " We report a case about a patient with 2 - up - long complaints of Uncontrollable movements in his right side of the body subsided after carotid revascularization."

Introduction:

2.     The main content of the introduction should highlight the background information, which is related to limb fibrillation and carotid revascularization. It is suggested to reduce some content that is not related to the topic of the article.

3.     The statements in the introduction do not have a good correlation and continuity with each other. It is recommended that the author modify the content of this part.

4.     The sentence in line 70-85 does not seem to be the focus of this study or the main background knowledge, so it is recommended to delete it.

Case presentation:

5.  The author described a patient with involuntary right limb movement that lasted for 2 months. In the description of the article, it seems that the patient did not know he had a right limb fibrillation before admission for examination. How did the author determine that the patient's limb fibrillation lasted for two months?

6.  In Figure1-3, marker arrows should be used to indicate the specific location of the vascular stenosis, so that the reader can easily identify it. In figure3, the left and right sides should be noted.

7.  After the implementation of Eversion endarterectomy of the left internal artery (l.135), is there any follow-up imaging examination to confirm the recanalization of the stenosis?

8.  The paragraph from line 145-149 does not seem to be helpful to the content of the article. It is suggested to delete it.

Discussion:

9.     The author did not explain the causes and mechanisms of carotid artery endarterectomy for the elimination of involuntary limb fibrillation, and did not cite relevant references to explain.

Conclusions:

10.  Conclusions should be the main findings of the research in the article. It is suggested to delete some sentences unrelated to the main content of the article, such as, “Carotid artery stenosis as one of the differential diagnoses of the causes of chorea is reported just recently”“A case of a 71-year-old male patient with the main complaints of 2-months-long uncontrollable movements in his right side of the body reducing rapidly after revascularization was demonstrated.” and “there are only 4 patients reported with severe extracranial carotid artery stenosis as the cause of hyperkinetic movements in Pauls Stradins Clinical University Hospital.”

Reviewer 2 Report

Even though the subject of the manuscript is quite interesting, some specific points may need to be clarified or improved significantly. 

1- Please explain more about how your case is different.

2- Conclusion: Please add more details about what we can learn from these cases. Why was your case report added to the medical literature? 

3- Explain the novelty of the paper and compare it with other case reports. 

Round 2

Reviewer 1 Report

1.文章中还有一些细微的语法错误,请仔细检查并仔细修改,或进行专业的英语润色。

2.作者引用一些文献来解释颈动脉内膜切除术消除不自主运动的原因和机制。我建议在讨论部分对此进行适度的描述。

3. 作者描述了一名持续2个月的右肢不自主运动患者。 患者是否有既往卒中史或其他导致肢体震颤的疾病史。如果没有,请在文章的适当位置进行解释。

Round 3

Reviewer 1 Report

The author revised the relevant content of the article according to my suggestion. I don't have any other comments or suggestions for this article.